# Global burden and epidemiological prediction of polycystic ovary syndrome from 1990 to 2019: A systematic analysis from the Global Burden of Disease Study 2019

Jiacheng Zhang[1], Yutian Zhu[1], Jiaheng Wang[2], Hangqi Hu[1], Yuxin Jin[1], Xin Mao[3], Haolin Zhang[1], Yang Ye[1]*, Xiyan Xin[1]*, Dong Li[1]*

1 Department of Traditional Chinese Medicine, Peking University Third Hospital, Beijing, China, 2 First Clinical School of Medicine, Shaanxi University of Chinese Medicine, Shaanxi, China, 3 Department of Radiology, Peking University Third Hospital, Beijing, China

\* yeyang89@126.com (YY); xinxiyan198234@163.com (XX); lidong6512@sina.com (DL)

## Abstract

### Objective

To comprehensively assess the global, regional and national burden of polycystic ovary syndrome (PCOS) in incidence, prevalence, and years lived with disability (DLYs) based on the Global Burden of Disease Study (GBD) 2019.

### Methods

This was a cross-sectional descriptive study. Data on PCOS incidence, prevalence, and DLYs from 1990 to 2019 were obtained from the GBD study 2019. According to the commonwealth income, WHO region, and the sociodemographic index, the estimates were demonstrated along with the estimated annual percentage change (EAPC). The EAPC data were analyzed by four levels of hierarchical clustering and displayed in the world map. The Autoregressive Integrated Moving Average (ARIMA) and Bayesian age-period-cohort (BAPC) model was used to predict the PCOS burden in the next 20 years.

### Results

From 1990 to 2019, the number of PCOS incidence in one year increased from 1.4 million in 1990 to 2.1 million in 2019 (54.3%). Only the EAPC estimates of incidence in the Region of the Americas decreased, and their aged-standardized incidence rate (ASIR) values were the highest in 1990 and 2019. There was no significant correlation between human development index (HDI) and EAPC. However, when HDI < 0.7, EAPC of incidence and prevalence was positively correlated with HDI, and when HDI > 0.7, EAPC of incidence and prevalence was negatively correlated with HDI. Countries with the middle level HDI have the highest increasing trend of ASIR and age-standardized prevalence rate (ASPR). The 10 to 19 years old group had the highest incidence counts of PCOS globally. Besides, the ARIMA and BAPC model showed the consistent increasing trend of the burden of PCOS.

**Data Availability Statement:** All relevant data are within the manuscript and its Supporting information files.

**Funding:** This study was supported by the National Natural Science Foundation of China (No. 82074193 and 82174151), Capital's Funds for Health Improvement and Research (No. 2020-2-40912 and 2022-2-4097), Cohort Construction Project of Peking University Third Hospital (No. BYSYDL2022013), and Special Fund of the Beijing Clinical Key Specialty Construction Program, P. R. China (2022) (No. BJZKBC0011). The funders had no role in study design, data collection and analysis, decision to publish, or preparation of the manuscript.

**Competing interests:** The authors have declared that no competing interests exist.

## Conclusion

In order to better promote the early diagnosis and treatment, expert consensus and diagnosis criteria should be formulated according to the characteristics of different ethnic groups or regions. It is necessary to emphasize the early screening and actively develop targeted drugs for PCOS.

## Introduction

Polycystic ovary syndrome (PCOS), characterized by chronic anovulation, hyperandrogenism, and insulin resistance, is one of the most common reproductive endocrinopathy affecting approximately 8% to 18% reproductive-aged women [1]. In addition to being an important cause of infertility, it is also a risk factor for many metabolic diseases, such as type 2 diabetes mellitus, cardiovascular continuum, and endometrial carcinoma [2, 3]. In 2019, 66 million people worldwide suffered from PCOS [4]. The incidence and prevalence rates of PCOS are discrepant according to age and regional differences. But in general, the prevalence and incidence rates of PCOS are continuously increasing [5]. By 2020, the economic burden of PCOS was assessed as 8 billion dollars per year [6, 7]. However, up to date, no specific drug has been approved by the US FDA nor the European Medicines Agency [8]. The prescriptions given to PCOS patients, such as metformin, letrozole, clomiphene, and oral contraceptives, are off-label and symptom-oriented [9]. In order to provide foundation for the government to make decision on medical resource allocation and underlie reference for clinical research, it is necessary to explore the epidemiological characteristics of PCOS comprehensively.

The Global Burden of Disease (GBD) study is a comprehensive research project that aims to quantify the impact of various diseases, injuries, and risk factors on global health [10]. Previous investigation found that age and region are key factors in the epidemiological manifestations of PCOS. In 2019, globally, the 40 to 44 age group represented the highest prevalence, the 15 to 19 age group represented the highest incidence, and the 25 to 29 age group showed the highest number of years lived with disability (YLD). The numerical analysis of age standardization demonstrated that high-income Asia-Pacific, Australasia, and Western Europe are the regions with the largest disease burden of PCOS [4]. A GBD study of PCOS within Europe found the prevalence rate in the Czech Republic was 460.6 (per 100000), while in Sweden it was only 34.1 (per 100000) [11]. These estimates indicated the possibility of discovering new etiological determinants of PCOS from population-geography-society characteristics. However, the current GBD research on PCOS is relatively limited. Only one study based on GBD 2019 comprehensively analyzed the data [4], but no more in-depth data processing was conducted, such as the estimated annual percentage change (EAPC), and its correlation with the age-standardized rate (ASR) and human development index (HDI). To date, there is no research to model and evaluate the future burden of PCOS.

Herein, we retrieved information from the GBD 2019 study and conducted a profundity analysis to comprehensively display the global, regional, and national burden of PCOS. In addition to analyzing the prevalence, incidence, and YLDs, we further evaluated EAPC and its heterogeneity sources. The Autoregressive Integrated Moving Average (ARIMA) and Bayesian age-period-cohort (BAPC) model was used to assess the burden of PCOS in the next 20 years.

## Materials and methods

### Data sources

Annual cases and percentages of PCOS from 1990 to 2019, by region and country were obtained from the Global Health Data Exchange (GHDx) query tool (https://vizhub. healthdata.org/gbd-results) [12]. The DisMod-MR 2.1, which is a Bayesian meta-regression tool, was used to model the incidence and prevalence. YLDs were calculated by the prevalence estimates multiplied by disability weights for mutually exclusive sequelae of diseases [13]. 95% uncertainty intervals (UIs) were included with the estimates. The characteristics of GBD study versions and detailed steps of using this database have been described in previous studies [12].

### Case definitions

PCOS is defined as a common endocrine and metabolic disorder among reproductive-aged women, characterized by the presence of polycystic ovaries, hyperandrogenism, and menstrual dysfunction. The diagnosis of PCOS is often made using the Rotterdam criteria, which require the presence of at least two of the following: oligo-ovulation or anovulation, hyperandrogenism (clinical or biochemical), and polycystic ovaries on ultrasound. The GBD database categorizes PCOS as a non-communicable disease and includes it in the broader category of gynecological diseases. In GBD 2019, the diagnosis of PCOS can be made by any of the following approaches: Rotterdam criteria, NIH criteria, and the Androgen Excess and PCOS Society definition [4, 14].

### Measures of burden

From 1990 to 2019, the prevalence and incidence of PCOS in 204 countries and territories were shown on the world map, and the global distribution of aged-standardized incidence rate (ASIR) and age-standardized prevalence rate (ASPR) in 2019 was plotted. EAPC is a metric for measuring trends, commonly used to describe the rate of increase or decrease of a specific variable over a certain period of time. EAPC was calculated and displayed by fitting the regression line to the natural logarithm of the rates [15]. Estimates were classified according to the commonwealth income, WHO region, and the sociodemographic index (SDI).

### Cluster analysis and correlation analysis

Hierarchical Clustering Analysis is a clustering algorithm used to partition a set of data into multiple distinct clusters. Unlike other clustering algorithms, such as K-means, Hierarchical Clustering Analysis does not require the number of clusters to be specified in advance. Instead, it merges similar samples into a cluster based on the distance between each pair of samples until all samples are merged. The results were represented using a dendrogram, in which each leaf node represents a sample, and each internal node represents the merging of clusters. The EAPC of incidence and prevalence of the 204 countries and territories were clustered into 4 categories (decrease or tiny increase, minor increase, stable increase, and significant increase).

Correlation analysis is a statistical method used to study the linear relationship between two variables by calculating the correlation coefficient. The correlation coefficient is a standardized measure that describes the strength and direction of the linear relationship between two variables. The cor.test function in R was used to calculate the correlation coefficients between EAPC, ASIR, ASPR, and HDI. The results were illustrated in the form of a scatter plot combined with a trend line.

### ARIMA model and BAPC model for forecasting (2020–2042)

The ARIMA model was applied to predict the prevalence and incidence rate of PCOS. The core of ARIMA is to difference the time series to transform the non-stationary series into a stationary one and then model the stationary series. The ARIMA model includes three main parameters: p, d, and q. Herein, p represents the order of the autoregressive term, d represents the order of differencing, and q represents the order of the moving average term. Autocorrelation function (ACF) and partial autocorrelation function (PACF) were used to determine the parameters. The *forecast* and *tseries* packages were used for ARIMA model forecasting and visualization.

In addition to the ARIMA model, the BAPC model was also applied to epidemiology prediction. This statistical model based on Bayesian statistical theory is used to analyze and explain the trends of individual attributes in the population with respect to age, period, and birth cohort. Unlike traditional classical statistical models, the BAPC model not only considers the influence of age and period but also models the birth cohort as a factor, providing a better understanding and description of the impact of birth cohort. Moreover, the BAPC model can also consider different prior distributions and hyperparameters, providing more flexible and personalized model selection and fitting methods. According to the standard age structure in GBD and the predicted population data of WHO, the prevalence and incidence of PCOS in the next 20 years were predicted with the BAPC model. The *BAPC* and *INLA* packages were used for BAPC model forecasting and visualization.

### Statistical analysis

Counts and ASR (per 100,000) of prevalence, incidence, and YLDs were used as the assessment of the burden of PCOS with 95% UIs. All statistical analyses and visualization were conducted by R (version 4.2.2). The detailed information of packages used in this study was listed in S1 Fig in S1 File. P-value < 0.05 was considered statistically significant.

## Results

### Global, regional and national level

From 1990 to 2019, there was an increase of 32 million PCOS patients worldwide. The incidence of PCOS in one year increased from 1.4 million in 1990 to 2.1 million in 2019. The YLDs increased from 0.3 million to 0.6 million. The EAPC estimates of prevalence, incidence, and YLDs were 0.84, 0.85, and 0.83, respectively (Fig 1).

From the WHO region classification, the region which had the highest incidence in 1990 was the Western Pacific Region, while in 2019, the South-East Asia Region showed the highest incidence. Although only the EAPC estimates of the Region of the Americas decreased, their ASIR values were the highest in 1990 and 2019. The EAPC of incidence estimates of South-East Asia region was the highest, and the ASIR of other regions did not decrease. Similar to the incidence data, EAPC of prevalence estimates of the Americas region was the only one that decreased, and the ASPR estimates were the highest. The counts of prevalence of Western Pacific Region were the highest in 1990 and 2019. The regional trends of YLDs were consistent with the incidence rate. ASR were the highest in Region of the Americas in 1990 and 2019, while the EAPC estimates were highest in South-East Asia Region (Fig 1).

From the region classification of SDI, the Middle SDI regions had the highest incidence counts in 1990 and 2019, while the High SDI regions showed the highest ASIR estimates in 1990 and 2019, only the incidence EAPC estimates in the High SDI regions were negative. In 1990, the High SDI regions had the highest number of prevalence, while in 2019, the counts of

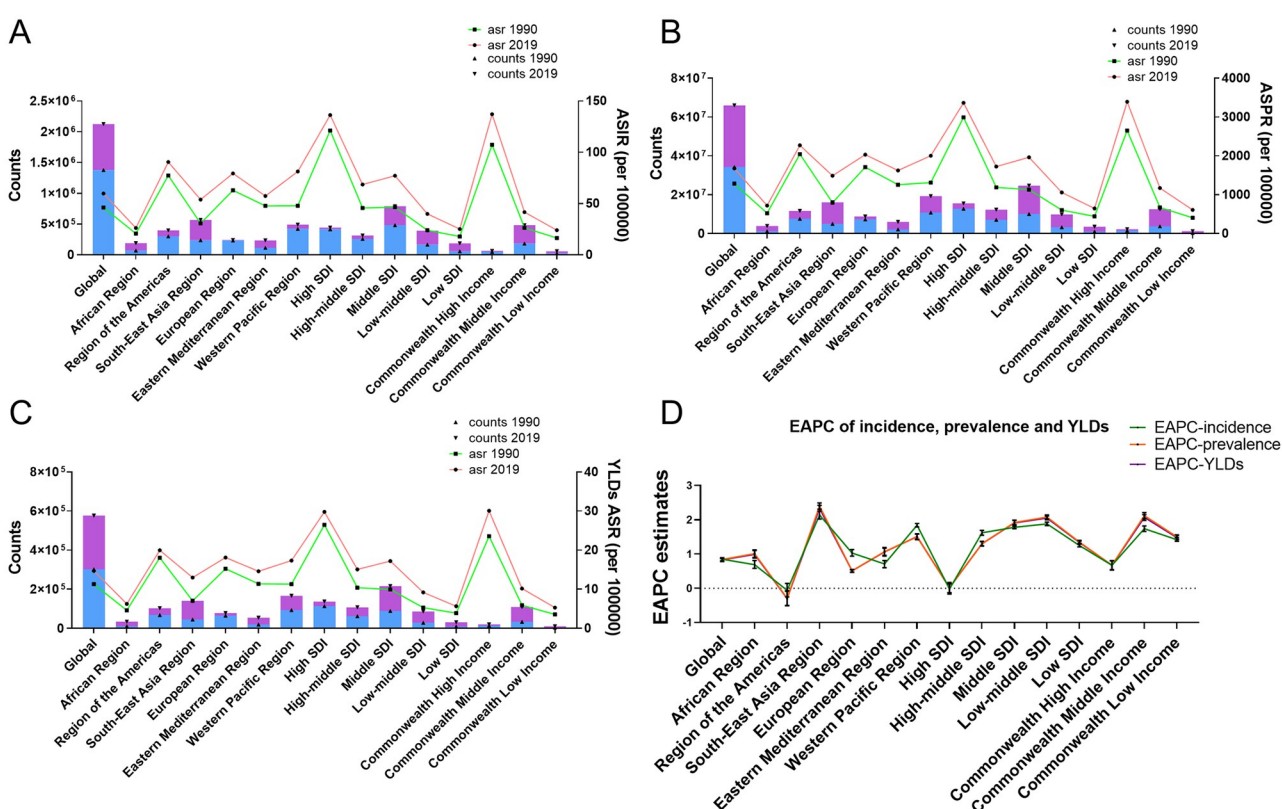

**Fig 1. Global and regional burden of PCOS from 1990 to 2019.** (A) global and regional estimates of ASIR from 1990 to 2019. (B) global and regional estimates of ASPR from 1990 to 2019. (C) global and regional estimates of YLDs ASR from 1990 to 2019. (D) global and regional EAPC of incidence, prevalence and YLDs.

prevalence in this region ranked second, and the Middle SDI regions had the highest amount of prevalence. The prevalence estimates of EAPC in each region classified according to SDI were positive, and the highest was the Low-middle SDI regions. In 1990, the estimated counts of YLDs were the highest in the High SDI regions, while in 2019, the highest counts were in the Middle SDI regions. EAPC estimates of YLDs were all positive, and the lowest were the High SDI regions (Fig 1).

From the region classification of Commonwealth Income, the counts of incidence, prevalence, and YLDs in the Commonwealth Middle Income region were the highest both in 1990 and 2019, while the ASR estimates of these three were the highest in the Commonwealth High Income region. The EAPC estimates of the three values were also the highest in the Commonwealth Middle Income region (Fig 1, Table 1).

In 2019, among the 204 countries and territories, Albania, Bosnia and Herzegovina, North Macedonia, and Serbia had the lowest ASIR, Italy, Japan, and New Zealand had the highest ASIR. Similar to the incidence data, these four countries with the lowest ASIR had the lowest ASPR. In addition to Italy, Japan, and New Zealand, Australia and Malaysia showed particularly high ASPR. The analysis of changes in prevalence from 1990 to 2019 revealed that Italy, Japan, Latvia, Bermuda, Bulgaria, Northern Mariana Islands, and Lithuania all decreased, and Equatorial Guinea, Qatar, United Arab Emirates and Maldives increased the most. Incidence of 34 countries and territories decreased, with the largest increased scale in Japan (-35%) and the largest decreased scale in Equatorial Guinea (749%). There were seven countries or

Table 1. Global and regional incidence, prevalence and YLDs of PCOS.

| location | Incidence | | | | | Prevalence | | | | | YLDs | | | | |
|---|---|---|---|---|---|---|---|---|---|---|---|---|---|---|---|
| | Num_1990 | ASR_1990 | Num_2019 | ASR_2019 | EAPC_CI | Num_1990 | ASR_1990 | Num_2019 | ASR_2019 | EAPC_CI | Num_1990 | ASR_1990 | Num_2019 | ASR_2019 | EAPC_CI |
| Global | 1377924.6 (941751–1816993.7) | 46.1 (31.6–61) | 2125511.8 (1489953.8–2803300.7) | 59.8 (41.7–78.9) | 0.85 (0.83–0.87) | 34263074 (23600055.2–45029362.6) | 1286.2 (889.3–1698.3) | 65992317.6 (46033196.2–86273109.6) | 1677.8 (1166–2192.4) | 0.84 (0.8–0.89) | 301987 (127370.3–606719) | 11.3 (4.8–22.7) | 576822 (246221.5–1161158.6) | 14.7 (6.3–29.5) | 0.83 (0.78–0.87) |
| High SDI | 442030.3 (327998.4–564015) | 121.3 (82.9–162.4) | 414696.5 (284569.2–562441.5) | 136.3 (100.6–172.3) | -0.02 (-0.17–0.13) | 12787455.8 (8783959.7–17032784.9) | 2990.4 (2045.1–3973.1) | 15552645.2 (11551742.6–19787755.4) | 3365.2 (2490.3–4264.1) | 0.02 (-0.13–0.17) | 113083.2 (48472.4–231334) | 26.5 (11.4–54.4) | 136959.5 (60261.6–276121.8) | 29.8 (13.1–59.7) | 0.02 (-0.13–0.16) |
| High-middle SDI | 314697.1 (216659.7–418076.3) | 45.6 (31.2–60.9) | 258756.3 (176987.6–343213.3) | 68.5 (46.5–91.5) | 1.62 (1.55–1.69) | 7114306.5 (4829172.8–9404585.8) | 1188.8 (805.3–1573.2) | 12318354 (8364116.1–16289610.1) | 1724.7 (1167.8–2288.9) | 1.3 (1.24–1.37) | 62264.1 (26156.1–125762.4) | 10.4 (4.4–21) | 107048.9 (45571.5–217650.6) | 15.1 (6.4–30.5) | 1.3 (1.23–1.36) |
| Middle SDI | 789535.2 (541584.8–1049799.6) | 46.5 (31.4–62.1) | 481044.4 (325116.4–641432.5) | 77.2 (52.7–103.2) | 1.78 (1.74–1.81) | 10080953.1 (6792914.4–13361897.3) | 1132.7 (764.1–1502.6) | 24675206.5 (16978218.8–32682456.9) | 1963.3 (1346.8–2606.3) | 1.92 (1.86–1.99) | 88925.4 (37318.5–180282.6) | 9.9 (4.2–20.2) | 215592.9 (92249.8–437657.8) | 17.2 (7.3–34.9) | 1.91 (1.84–1.98) |
| Low-middle SDI | 392085.4 (260976.1–527328) | 23.9 (15.9–32) | 165167.6 (109661.3–221466.6) | 39.9 (26.6–53.8) | 1.88 (1.83–1.93) | 3222094 (2131496.7–4360538.8) | 603.1 (401.5–815.9) | 9883350.7 (6619036.4–13237117.6) | 1058.1 (710.9–1417.6) | 2.08 (2.02–2.14) | 28399.1 (12027.1–57275.1) | 5.3 (2.2–10.7) | 86016.9 (36623.7–174144.2) | 9.2 (3.9–18.7) | 2.05 (2–2.11) |
| Low SDI | 185659.8 (120988–253847.9) | 17.9 (11.7–24) | 57413.5 (36760.7–78232.9) | 25.1 (16.6–34) | 1.25 (1.21–1.29) | 1037808.2 (657166.3–1427276.9) | 444.5 (285–609.4) | 3518984.1 (2273812–4813593.1) | 646.7 (422–882.5) | 1.35 (1.31–1.4) | 9132.8 (3815.2–18906.9) | 3.9 (1.6–8) | 30816.6 (12883.1–62669.6) | 5.6 (2.4–11.5) | 1.34 (1.3–1.38) |
| Commonwealth High Income | 63537.3 (43565.6–84501.8) | 107.3 (75.1–141.3) | 47540.2 (33251.4–62868.5) | 137.1 (93.5–182.1) | 0.67 (0.54–0.8) | 1532581 (1082455.4–2021438.8) | 2651.9 (1870.8–3488.7) | 2246521.5 (1540821.7–2966661.1) | 3393.6 (2336.1–4459) | 0.67 (0.54–0.8) | 13601 (5778.1–27779.4) | 23.6 (10–48) | 19830.1 (8453–39629.8) | 30.1 (12.8–59.5) | 0.67 (0.54–0.8) |
| Commonwealth Middle Income | 481741.3 (323335.5–644674.4) | 26.3 (17.8–35) | 182846.1 (122588.6–245424.6) | 41.6 (28–55.7) | 1.74 (1.66–1.82) | 3727394.4 (2492503.4–5017637.2) | 672.5 (450.1–897.3) | 12528161.2 (8427716–16709558.4) | 1171.4 (788.7–1562.7) | 2.12 (2.03–2.21) | 32927.7 (13736.6–66244.6) | 5.9 (2.5–11.8) | 108838.1 (45995–219895.9) | 10.2 (4.3–20.5) | 2.07 (1.98–2.15) |
| Commonwealth Low Income | 57718.1 (37140.9–79112.5) | 16.3 (10.4–22.6) | 22248.2 (13911.7–31094.8) | 24 (15.7–32.9) | 1.42 (1.37–1.48) | 385394.1 (238022.5–544965.1) | 406.9 (256–570.4) | 1212139.6 (770862.8–1675141.9) | 609.9 (389.3–840.1) | 1.5 (1.43–1.56) | 3399.4 (1401.3–7047.1) | 3.5 (1.5–7.3) | 10539.3 (4340.8–21642.8) | 5.3 (2.2–10.9) | 1.48 (1.42–1.54) |
| African Region | 189240.3 (122716.8–259141.6) | 20.6 (13.5–27.9) | 67804.8 (43859.1–92376.7) | 26.2 (17.2–35.4) | 0.69 (0.58–0.81) | 1219774.8 (781302.6–1680411.7) | 525 (338.3–717.2) | 3896191 (2530243.9–5361078.7) | 720.1 (471.9–986.9) | 1.01 (0.9–1.12) | 10732 (4497.3–22039.6) | 4.6 (1.9–9.3) | 34011.5 (14202.6–69726.3) | 6.2 (2.6–12.8) | 0.99 (0.88–1.1) |
| Eastern Mediterranean Region | 234326.7 (156370.8–316608.4) | 47.8 (32.2–63.8) | 114044.2 (76053–153685.6) | 57.3 (38.6–77.3) | 0.71 (0.6–0.81) | 2126364.1 (1423182.5–2835729.7) | 1257.1 (844.6–1672.1) | 6013502.1 (4028195.9–8098161.4) | 1624.4 (1088.5–2189.9) | 1.07 (0.96–1.19) | 19395.5 (8231.6–38996.8) | 11.4 (4.8–23.1) | 54114.1 (22788–110879.1) | 14.6 (6.1–29.9) | 1.06 (0.94–1.17) |
| European Region | 241347.6 (164097.1–323025.8) | 62.9 (42.4–84.9) | 232911.1 (158080.7–313179.3) | 79.4 (53.8–106.9) | 1.03 (0.94–1.13) | 7322576.8 (5003378.6–9688992) | 1709.2 (1162.2–2260.3) | 8816298 (6047627.9–11752479.9) | 2030.6 (1385.8–2710) | 0.51 (0.47–0.55) | 65265.2 (27838.1–130485.3) | 15.2 (6.5–30.5) | 78142.8 (32931.2–157585) | 18.1 (7.7–36.4) | 0.51 (0.47–0.54) |
| Region of the Americas | 395889.1 (300257.1–501247.3) | 77.3 (53.6–102.9) | 300302.9 (207993.5–400221.9) | 90.5 (68.2–114.7) | -0.06 (-0.25–0.14) | 7630324.5 (5294503.8–10080562.4) | 2044.3 (1418.6–2708) | 11691625.8 (8840451.3–14663526.3) | 2271.8 (1719.7–2858.1) | -0.29 (-0.5–0.07) | 67528.9 (28743.2–136969.1) | 18.1 (7.7–36.6) | 102596.6 (45016–204140.2) | 20 (8.8–39.9) | -0.3 (-0.51–0.09) |
| South-East Asia Region | 567366.9 (385028.7–757185.4) | 30.7 (20.6–40.8) | 237472.3 (158830.6–319180.1) | 53.8 (36.4–72.2) | 2.14 (2.02–2.25) | 5041962.6 (3374228.9–6829457.2) | 794.6 (533.1–1069.1) | 16090267.8 (10926074–21384764.1) | 1488.7 (1011.7–1980.5) | 2.4 (2.31–2.49) | 44868.9 (18920.9–90844.5) | 7 (3–14.2) | 140219.9 (60193.9–283086.4) | 13 (5.6–26.2) | 2.33 (2.25–2.42) |
| Western Pacific Region | 491057.4 (338885.6–649878.7) | 47.9 (32.3–63.9) | 421252.4 (285906.1–565714.2) | 81.3 (55.9–108.1) | 1.84 (1.79–1.89) | 10813605.9 (7252025.3–14497100.2) | 1312.3 (881.7–1756.8) | 19272557.3 (13271424.4–25537807.3) | 2004.8 (1381.1–2642.8) | 1.5 (1.42–1.58) | 93244.9 (39567.5–191803.2) | 11.3 (4.8–23.3) | 165880.6 (71075.5–335469.7) | 17.3 (7.4–35.1) | 1.51 (1.44–1.59) |

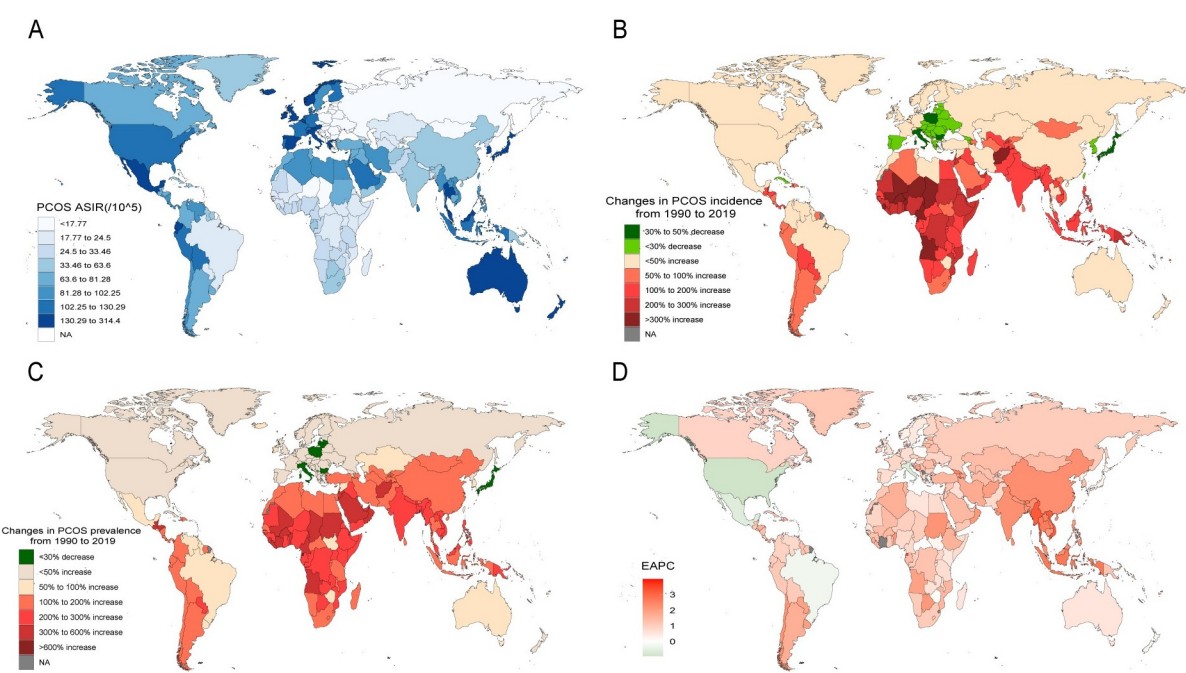

**Fig 2. National burden of PCOS.** (A) estimates of ASIR in 2019. (B) changes of incidence counts from 1990 to 2019. (C) changes of prevalence counts from 1990 to 2019. (D) national EAPC of incidence.

territories with negative EAPC estimates, among which the United States of America had the lowest. Zimbabwe and the United States of America were the only two countries with increased prevalence and negative EAPC estimates (Fig 2, S2 Fig in S1 File).

Hierarchical clustering of incidence and prevalence EAPC showed that 17 countries were divided into decrease or tiny increase group, 4 countries were divided into minor increase group, 117 countries were divided into stable increase group, and 66 countries were divided into significant increase group (Fig 3).

## Age pattern

In the GBD study, the age groups are divided into 5 years. There are 21 groups from 0 to 100 plus years old. The incidence and prevalence of PCOS were only recorded between 10 to 54 years old, which can be divided into nine groups. Globally, the 10 to 14 group and the 15 to 19 group had the highest incidence counts of PCOS, and the incidence of the later age groups declined precipitously. The prevalence data showed that the counts peaked in the 20 to 24 group in 1990, while peaked in the 25 to 29 group in 2019. From 1990 to 2019, the largest difference in the prevalence changes was in the 30 to 34 group. According to the analysis of the Age-WHO region classification, only the incidence of the 10 to 14 group in the European Region and the Eastern Mediterranean Region was larger than that of the 15 to 19 group. In 2019, the prevalence of the most regions peaked in the 20 to 29 group, while only the Western Pacific Region peaked in the 30 to 34 group, and the European Region peaked in the 40 to 44 group. According to the Age-SDI region classification, the incidence of the High-middle SDI region and the Middle SDI region was higher in the 10 to 14 group than in the 15 to 19 group. The peak age groups of the Low-middle SDI region and the Low SDI region were the 20 to 24 group, while the peak age groups of the High SDI region, the High-middle SDI region and the

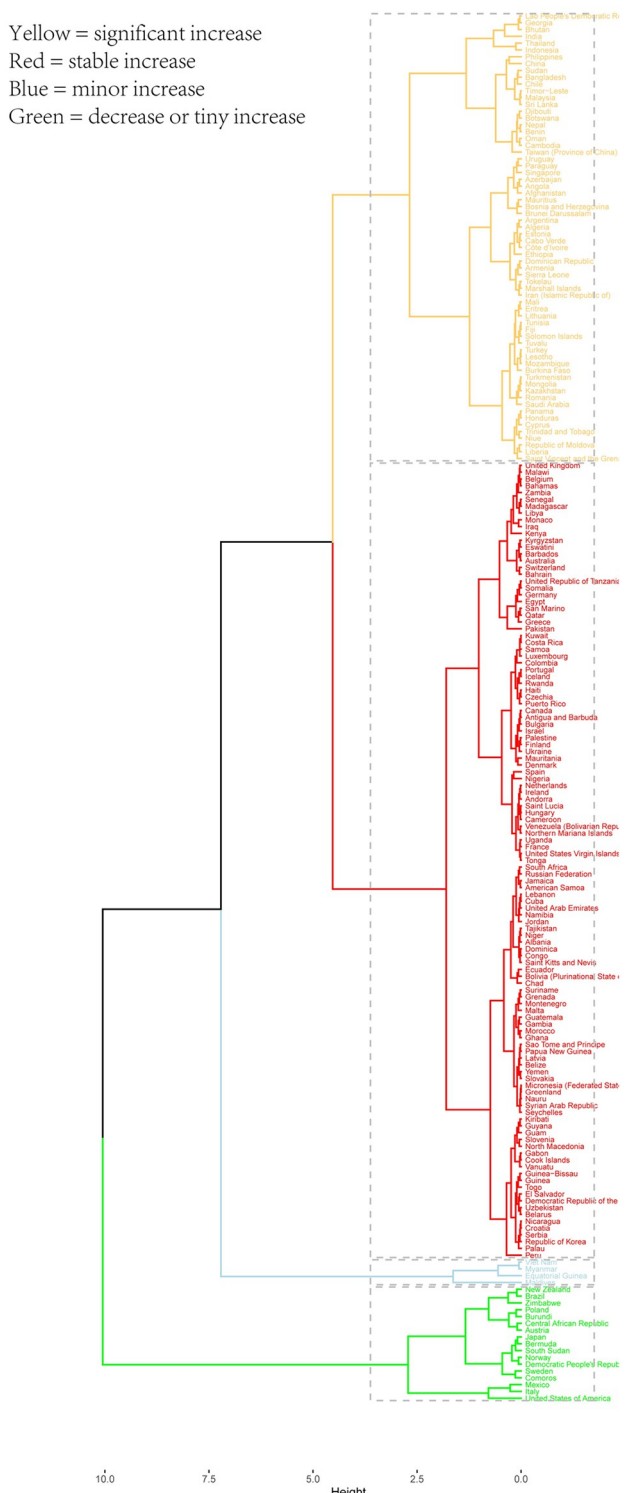

**Fig 3. Hierarchical clustering of 204 countries.**

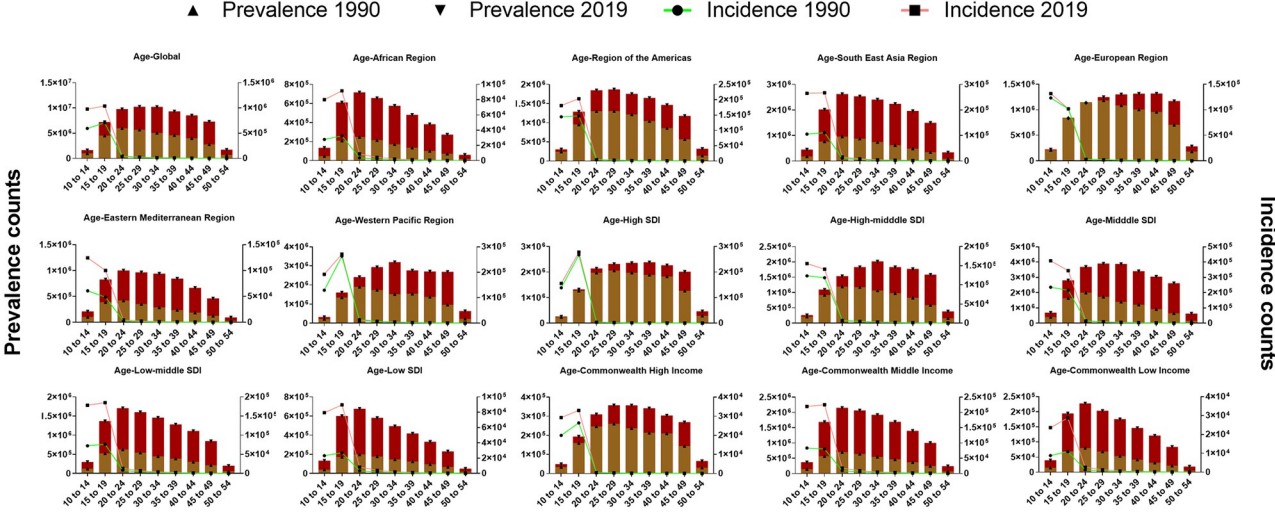

**Fig 4. Analysis of incidence and prevalence in age-region groups from 1990 to 2019.**

Middle SDI region are older. According to the Age-Commonwealth Income classification, the trends of incidence and prevalence of these three were similar. The peak age group of prevalence is the 25 to 29 group, while in the other two regions, the 20 to 24 group peaked (Fig 4, S1 Table in S1 File).

## Annual pattern

Globally, from 1990 to 2019, the ASPR and ASIR time series data of PCOS showed an increasing trend year by year, and the changing trend of ASPR and ASIR in each region was consistent. According to the classification of WHO region, only the incidence of the African Region decreased over a period of time (from 2007 to 2010), and the prevalence and incidence of the Region of the Americas decreased over a period of time (from 1999 to 2010). According to the SDI region classification, only the prevalence and incidence of the High SDI region did not increase annually [from 2001 to 2010 (both in prevalence and incidence)]. The regions classified by Commonwealth Income maintained a similar increasing trend (Fig 5, S2 Table in S1 File).

## EAPC influential factors

As is shown in Fig 6, the ASR in 1990 was regarded as the baseline disease data of PCOS, the relation between ASIR and ASPR with EAPC was negatively correlated with statistical differences (t = -5.3, p<0.01 in ASIR, t = -5.4, p<0.01 in ASPR). The HDI of each country was regarded as a sign of health care level, according to the distribution of HDI, there was no significant correlation between HDI and EAPC (t = -1.7, p = 0.09 in ASIR, t = -2.0, p = 0.05 in ASPR). However, when HDI < 0.7, EAPC of incidence and prevalence was positively correlated with HDI, and when HDI > 0.7, EAPC of incidence and prevalence was negatively correlated with HDI. Countries with the middle level HDI have the highest increasing trend of ASIR and ASPR (S3 and S4 Tables in S1 File).

## Prediction of the global PCOS burden

The ARIMA models of ASPR and ASIR demonstrated that the data become stationary only after three differencing operations (p = 0.01) (S3 Fig in S1 File). The ACF plot of ASPR

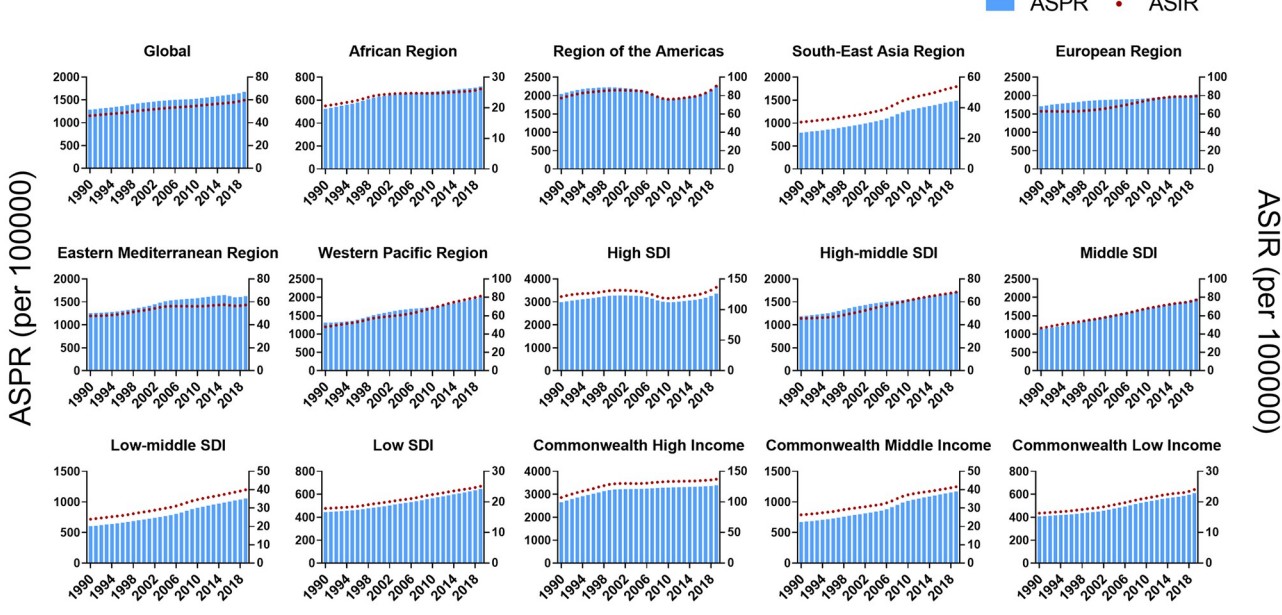

**Fig 5. Analysis of incidence and prevalence in time series-region groups from 1990 to 2019.**

indicated that the autocorrelation values exceed the boundary at the third lag, and the PACF plot showed that the partial autocorrelation values become stationary after the third lag. Therefore, an ARMA (3,3) model was selected. The ACF plot of ASIR showed that the autocorrelation values exceed the boundary at the third lag, and the PACF plot showed that the partial autocorrelation values become stationary after the second lag. Therefore, an ARMA (3,2) model was selected. The predicted results of the ARIMA model (S5 Table in S1 File) showed that by 2042, the ASIR and ASPR of PCOS will reach 112 (per 100000) and 3250 (per 100000), respectively. It should be noted that the series with high volatility requiring three differencing operations need to be carefully evaluated when interpreting the predicted results.

The prediction of BAPC model (Fig 7) revealed that the prevalence and incidence of PCOS would consistently increase in the next 20 years. In 2042, the predicted ASPR was 3806 (per 100000). According to the age group classification, the 35 to 39 group showed the highest prevalence of 5337 (per 100000). The predicted ASIR in 2042 was 117 (per 100000). According to the age groups, the incidence rate of the 10 to 14 group was 433 (per 100000), and the 15 to 19 group was 469 (per 100000) (S6 and S7 Tables in S1 File).

## Discussion

Compared with the other publications summarizing the global burden of PCOS based on the GBD study 2019, this study added data analysis of prevalence and incidence by multiple regions and countries, age groups, and time series to explore the heterogenous sources of estimates. The Region of the Americas showed the highest ASIR and ASPR in 1990 and 2019, but the South-East Asia Region and the Western Pacific Region had the highest counts of incidence and prevalence, which was related to the total population of the regions. The least ASIR and ASPR were shown in the African Region. The other article, different from the region classification of this study, claimed that High-income Asia-Pacific, Australia, and Western Europe were the regions with the heaviest disease burden [4]. The race and ethnicity differences were

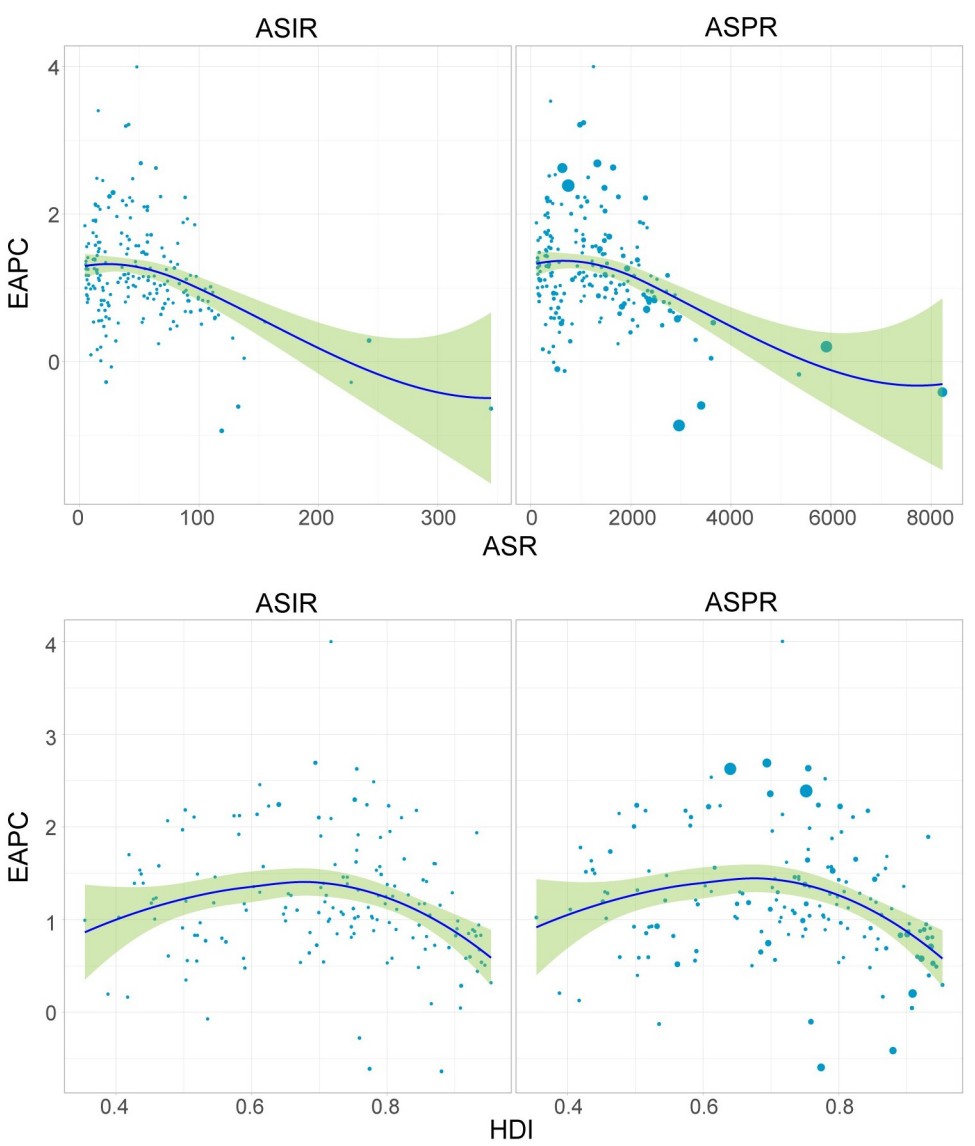

**Fig 6. Correlation analysis of EAPC with (A) ASIR and ASPR, and (B) HDI.**

first considered the sources of heterogeneity. Analyses of the genomic databases revealed that the ethnic variability of PCOS was indeed determined by the human genetic background, and ethnic variations in PCOS phenotypic expression occur in all regions [16, 17]. For instance, body mass index (BMI) and homeostasis model assessment-insulin resistance (HOMA-IR) of Mexican-American women are higher than those of Caucasian women, and the hairiness of Asian women is significantly lower than that of Caucasian women [18]. These symptoms are part of the diagnostic criteria of PCOS, so racial phenotypic variations largely determine the different characteristics in disease.

Different diagnostic criteria may lead to changes in epidemiological data. In the time series analysis, the ASIR of PCOS in Region of the Americas decreased from 1999 to 2010, which was not consistent with the prediction after the emergence of the Rotterdam diagnostic criteria. The Rotterdam criteria with two new phenotypes should have extended the definition of

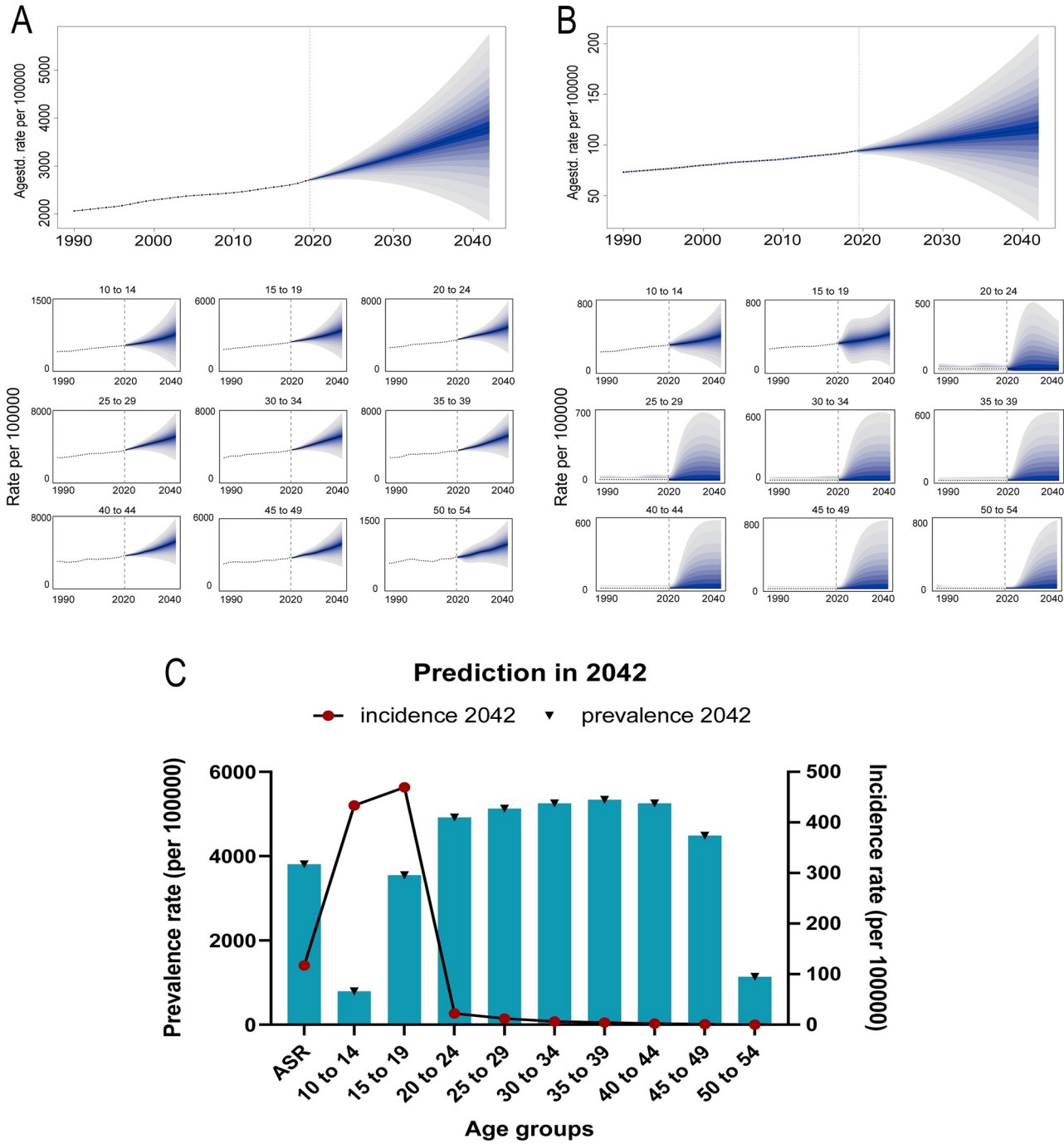

**Fig 7. BAPC prediction of PCOS burden in the next 20 years.** (A) ASIR and incidence rate of age groups from 1990 to 2042. (B) ASPR and prevalence rate of age groups from 1990 to 2042. (C) Estimates of ASIR and ASPR in 2042.

PCOS, and the prevalence ought to increase contrary to the existing decreasing trend. In addition to the international diagnostic criteria, some countries and territories have developed unique PCOS criteria based on the characteristics of the population. PCOS diagnostic criteria formulated by the Japanese Society of Obstetrics and Gynecology (JSOG) in 1993 and updated

in 2007 [19, 20]. The JSOG criteria proposed the importance of LH and FSH in diagnosing PCOS, which is more suitable for the characteristics of people in eastern Asia. This may partly explain why Japan showed the highest ASIR and ASPR among the 204 countries and territories.

Previous studies have shown that there is a positive correlation between SDI and the burden of PCOS, which may be because the westernized diet is more popular in developed countries, which is closely related to the risk factors of PCOS, such as obesity and insulin resistance [4]. In this study, SDIs were divided into five grades, although the ASPR of the High SDI region was the highest in 1990, the ASIR and ASPR of the Middle SDI region were both the highest in 2019. If the regions were classified into three categories according to commonwealth income, the ASIR and ASPR of the middle level regions were the highest. Therefore, it is inaccurate to simply conclude that high SDI represents a high PCOS burden. In order to further explore the correlation between the two, EAPC and HDI were applied to the correlation analysis. HDI is composed of three indicators: life expectancy, adult literacy rate, and logarithm of GDP per capita [21, 22]. Our results demonstrated that when HDI was < 0.7, EAPC and HDI were positively related, while in the case of HDI > 0.7, the relationship between them was reversed. First of all, compared with the low HDI regions, the average life expectancy of residents in the middle HDI regions has increased more [23]. As a chronic disease, PCOS patients have a longer survival period, which may lead to an increase in prevalence. In the process of population aging, industrialization, and urbanization, inevitably deteriorating social and environmental factors also have a significant impact on the incidence of PCOS. Individuals in the middle HDI regions may maintain higher behavioral risk factors, such as unsuitable eating habits leading to excessive nutrition, carbohydrate, and fat intake exceeding consumption for a long time [24, 25]. Poor living habits and lack of exercise are also risk factors for PCOS and metabolic diseases [26].

From the perspective of age pattern, the 15 to 19 group had the highest incidence of PCOS. Only in the European Region and the Eastern Mediterranean Region, this contrast was reversed. Because PCOS is a hormone-related disease, which mainly occurs in women of childbearing age, this may be explained by the characteristics of the age of sexual maturity in different regions [27]. From the prevalence analysis, the concentration of PCOS age groups has changed from the 20 to 24 group in 1990 to the 25 to 19 group in 2019 globally, which can be explained by the mentioned increase in life expectancy and population aging. Prevalence of PCOS in most regions peaked in the 20 to 29 group and declined slowly, which is related to the active treatment and the age-incidence rate of PCOS. The European Region showed a different age-prevalence pattern in 2019. The counts of PCOS patients increased steadily from the age group of 10 to 14 to the 40 to 44 group. This was closely related to the wave of European immigrants and the formation of an aging society. According to the Eurostat data in 2019, the population aged 65 and over in 27 EU countries reached 90.5 million, accounting for 20.3% of the total population [28].

This study has the following limitations. Firstly, the DisModMR model of GBD only considers the NIH criteria as the reference standard for diagnosing PCOS, ignoring some local definitions, which may lead to heterogeneity in epidemiological data. In particular, the NIH criteria rely heavily on the presence of oligo-ovulation and hyperandrogenism, which may not be suitable for some populations. As a result, the prevalence of PCOS may be underestimated or overestimated, which affects the accuracy of burden estimates. Secondly, when GBD collected and collated regional data, it was inevitable that the regional data was not completely obtained, and it was difficult to implement regional analysis within a country. For example, some studies may not provide regional data, while others may provide data that is not consistent with the GBD's regional classification. Therefore, the regional burden of PCOS may not

be accurately estimated, which affects the effectiveness of targeted intervention and resource allocation. Finally, the risk factors of PCOS were not found in the GBD study database. PCOS is a multifactorial disorder, and many factors have been implicated in its pathogenesis, including genetic, environmental, and lifestyle factors. The study may not provide a comprehensive understanding of the risk factors of PCOS, which hinders the development of effective prevention and control strategies.

## Conclusion

This study revealed that the incidence, prevalence, and YLDs of PCOS were increasing and this trend would maintain in the next 20 years. The social and economic development is not fully positively related to the PCOS burden, and the burden is highest in medium regions. Women's health infrastructure should be strengthened to deal with potential PCOS patients in the future. The highest incidence of PCOS is from 10 to 19 years old, which points out that the government should pay attention to the importance of early screening in adolescents.

## Supporting information

**S1 File.**
(PDF)

## Author Contributions

**Conceptualization:** Yang Ye, Dong Li.

**Data curation:** Jiacheng Zhang, Yutian Zhu.

**Formal analysis:** Jiacheng Zhang, Jiaheng Wang.

**Funding acquisition:** Yang Ye, Dong Li.

**Investigation:** Jiacheng Zhang, Hangqi Hu, Yuxin Jin, Haolin Zhang.

**Methodology:** Xin Mao, Xiyan Xin.

**Project administration:** Yang Ye, Dong Li.

**Software:** Jiacheng Zhang, Yutian Zhu.

**Supervision:** Yang Ye, Dong Li.

**Visualization:** Jiacheng Zhang, Yutian Zhu.

**Writing – original draft:** Jiacheng Zhang, Yutian Zhu.

**Writing – review & editing:** Yang Ye, Dong Li.

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
