## [Decision Letter · Decision Letter 0]

27 May 2024

PONE-D-24-15043Global burden and epidemiological prediction of polycystic ovary syndrome from 1990 to 2019: a systematic analysis from the Global Burden of Disease Study 2019PLOS ONE

Dear Dr. Ye,

Thank you for submitting your manuscript to PLOS ONE. After careful consideration, we feel that it has merit but does not fully meet PLOS ONE’s publication criteria as it currently stands. Therefore, we invite you to submit a revised version of the manuscript that addresses the points raised during the review process.

Please submit your revised manuscript by Jul 11 2024 11:59PM. If you will need more time than this to complete your revisions, please reply to this message or contact the journal office at plosone@plos.org. Please include the following items when submitting your revised manuscript:A rebuttal letter that responds to each point raised by the academic editor and reviewer(s). You should upload this letter as a separate file labeled 'Response to Reviewers'.A marked-up copy of your manuscript that highlights changes made to the original version. You should upload this as a separate file labeled 'Revised Manuscript with Track Changes'.An unmarked version of your revised paper without tracked changes. You should upload this as a separate file labeled 'Manuscript'.If applicable, we recommend that you deposit your laboratory protocols in protocols.io to enhance the reproducibility of your results. Protocols.io assigns your protocol its own identifier (DOI) so that it can be cited independently in the future. For instructions see: https://journals.plos.org/plosone/s/submission-guidelines#loc-laboratory-protocols. Additionally, PLOS ONE offers an option for publishing peer-reviewed Lab Protocol articles, which describe protocols hosted on protocols.io. Read more information on sharing protocols at https://plos.org/protocols?utm_medium=editorial-email&utm_source=authorletters&utm_campaign=protocols.

We look forward to receiving your revised manuscript.

Kind regards,

Zhaoqing Du, Ph.D

Academic Editor

PLOS ONE

Journal Requirements:

"This study was funded by the National Natural Science Foundation of China (No. 82074193 and 82174151) and the Special Grant for Capital Health Research and Development (No. 2020-2-40912, 2022-2-4097, and 2022-2-4098)."

4. We note that [Figures 2 and Fig S2] in your submission contain [map/satellite] images which may be copyrighted. All PLOS content is published under the Creative Commons Attribution License (CC BY 4.0), which means that the manuscript, images, and Supporting Information files will be freely available online, and any third party is permitted to access, download, copy, distribute, and use these materials in any way, even commercially, with proper attribution. For these reasons, we cannot publish previously copyrighted maps or satellite images created using proprietary data, such as Google software (Google Maps, Street View, and Earth). For more information, see our copyright guidelines: http://journals.plos.org/plosone/s/licenses-and-copyright.

a. You may seek permission from the original copyright holder of Figures 2 and Fig S2 to publish the content specifically under the CC BY 4.0 license.  

Reviewers' comments:

Reviewer's Responses to Questions

**Comments to the Author**

1. Is the manuscript technically sound, and do the data support the conclusions?

Reviewer #1: Yes

Reviewer #2: Yes

Reviewer #3: Yes

2. Has the statistical analysis been performed appropriately and rigorously? 

Reviewer #1: Yes

Reviewer #2: Yes

Reviewer #3: Yes

3. Have the authors made all data underlying the findings in their manuscript fully available?

Reviewer #1: Yes

Reviewer #2: Yes

Reviewer #3: Yes

4. Is the manuscript presented in an intelligible fashion and written in standard English?

Reviewer #1: Yes

Reviewer #2: No

Reviewer #3: Yes

5. Review Comments to the Author

Reviewer #1: The manuscript is well-organized, presenting the research findings in a clear and understandable manner. The language used throughout is articulate and clear of any discernible spelling or grammatical errors. The conclusion drawn by the authors is both reasonable and well-supported by the comprehensive data provided.

Reviewer #2: Dear authors,

Thank you for submitting your work based on the Global Burden of Disease (2019) about the assessment of global, regional and national burden of PCOS in terms of incidence, prevalence, and years lived with disability (DLYs).

PCOS is a widespread disease affecting women during their reproductive years and beyond with its metabolic and oncological complications. It is an important health problem that should be addressed and concrete strategies drawn up.

I thank the authors for this article which could benefit the current efforts made globally, regionally, and nationally in order to advance PCOS management.

The manuscript is technically sound, and the methods permitted a thorough statistical analysis of the data.

However, I have few observations:

1. In the method section the correlation coefficient used was not mentioned.

2. The manuscript could benefit from editing. It must be clearer, with no ambiguity, which was the case for many sentences in the result sections.

148: There was an increase of 32 million PCOS patients worldwide…

149: The incidence of PCOS in one year increased from 1.4 million in 1990 to 2.1 million in 2019.

152: the region which had the highest incidence in 1990 ……..was the western pacific…..

154: this sentence should be rephrased so that we know that EAPC of incidence estimates of the Americas region was only one that decreased.

160: ASR were the highest in Region of the Americas in 1990 and 2019, while the EAPC estimates were highest in South-East Asia Region.

3. The legends in all the figures were illegible, so it was impossible to read them.

Thank you for your work.

Reviewer #3: This paper is, somehow, an interesting excercise of probabilistic (Bayesian) approaches to a previously collected data (from the Global Health data Exchange) . It is, as a hole, a very neat metodological excercise. From the clinical point of view, as it should be expected, it does not add any important data.

Methodologically, it is a very well designed analysis and allows to withdraw some info which, considering the heterogenicity of the sample and the variety of criteria involved, renders results that should be taken precauciously.

From the stylistic point of view, there are several abbreviations in the abstract that are totally inintelligible unless you read the entire paper (ASIR,HDI,ASIR and ASPIR). Those should be, either, properly adressed or retired.

6. PLOS authors have the option to publish the peer review history of their article (what does this mean?). If published, this will include your full peer review and any attached files.

Reviewer #1: **Yes: **Lara Abu-Qutaish

Reviewer #2: No

Reviewer #3: **Yes: **Juan Carlos Bello Munoz

---

## [Author Response · Author response to Decision Letter 0]

11 Jun 2024

Response to Reviewers

Reviewer #1: 

The manuscript is well-organized, presenting the research findings in a clear and understandable manner. The language used throughout is articulate and clear of any discernible spelling or grammatical errors. The conclusion drawn by the authors is both reasonable and well-supported by the comprehensive data provided.

Response: Thank you for recognizing our work, this is greatly helpful to us.

Reviewer #2: 

Dear authors,

Thank you for submitting your work based on the Global Burden of Disease (2019) about the assessment of global, regional and national burden of PCOS in terms of incidence, prevalence, and years lived with disability (DLYs).

PCOS is a widespread disease affecting women during their reproductive years and beyond with its metabolic and oncological complications. It is an important health problem that should be addressed and concrete strategies drawn up.

I thank the authors for this article which could benefit the current efforts made globally, regionally, and nationally in order to advance PCOS management.

The manuscript is technically sound, and the methods permitted a thorough statistical analysis of the data.

However, I have few observations:

1. In the method section the correlation coefficient used was not mentioned.

Response: Thank you for pointing out the issue. We have added the calculation method for the correlation coefficient in the “Cluster analysis and correlation analysis” section.

2. The manuscript could benefit from editing. It must be clearer, with no ambiguity, which was the case for many sentences in the result sections.

148: There was an increase of 32 million PCOS patients worldwide…

149: The incidence of PCOS in one year increased from 1.4 million in 1990 to 2.1 million in 2019.

152: the region which had the highest incidence in 1990 ……..was the western pacific…..

154: this sentence should be rephrased so that we know that EAPC of incidence estimates of the Americas region was only one that decreased.

160: ASR were the highest in Region of the Americas in 1990 and 2019, while the EAPC estimates were highest in South-East Asia Region.

Response: Thank you very much for pointing out the grammatical issues. This has made the manuscript more fluent, and we have made the corresponding corrections as per your suggestions.

3. The legends in all the figures were illegible, so it was impossible to read them.

Response: We have redrawn all the images with fonts that were too small or potentially non-standard and uploaded high-resolution versions. 

Thank you again for pointing out these issues; this has greatly improved the readability of our article.

Reviewer #3: 

This paper is, somehow, an interesting excercise of probabilistic (Bayesian) approaches to a previously collected data (from the Global Health data Exchange) . It is, as a hole, a very neat metodological excercise. From the clinical point of view, as it should be expected, it does not add any important data.

Methodologically, it is a very well designed analysis and allows to withdraw some info which, considering the heterogenicity of the sample and the variety of criteria involved, renders results that should be taken precauciously.

From the stylistic point of view, there are several abbreviations in the abstract that are totally inintelligible unless you read the entire paper (ASIR, HDI, ASIR and ASPIR). Those should be, either, properly adressed or retired.

Response: Thank you for pointing out the errors. We have added the full names for PCOS, ASIR, ASPR, and HDI in the abstract section.

---

## [Decision Letter · Decision Letter 1]

26 Jun 2024

Global burden and epidemiological prediction of polycystic ovary syndrome from 1990 to 2019: a systematic analysis from the Global Burden of Disease Study 2019

PONE-D-24-15043R1

Dear Dr. Yang Ye,

We’re pleased to inform you that your manuscript has been judged scientifically suitable for publication and will be formally accepted for publication once it meets all outstanding technical requirements.

Kind regards,

Zhaoqing Du, Ph.D

Academic Editor

PLOS ONE

Additional Editor Comments (optional):

Reviewers' comments:

Reviewer's Responses to Questions

**Comments to the Author**

1. If the authors have adequately addressed your comments raised in a previous round of review and you feel that this manuscript is now acceptable for publication, you may indicate that here to bypass the “Comments to the Author” section, enter your conflict of interest statement in the “Confidential to Editor” section, and submit your "Accept" recommendation.

Reviewer #2: All comments have been addressed

Reviewer #3: All comments have been addressed

2. Is the manuscript technically sound, and do the data support the conclusions?

Reviewer #2: Yes

Reviewer #3: Yes

3. Has the statistical analysis been performed appropriately and rigorously? 

Reviewer #2: Yes

Reviewer #3: Yes

4. Have the authors made all data underlying the findings in their manuscript fully available?

Reviewer #2: Yes

Reviewer #3: Yes

5. Is the manuscript presented in an intelligible fashion and written in standard English?

Reviewer #2: Yes

Reviewer #3: Yes

6. Review Comments to the Author

Reviewer #2: (No Response)

Reviewer #3: After thorough reading, I see no drawbacks for this version of the paper. All the suggested adjustments have been made

7. PLOS authors have the option to publish the peer review history of their article (what does this mean?). If published, this will include your full peer review and any attached files.

Reviewer #2: No

Reviewer #3: **Yes: **Juan Carlos BELLO MUÑOZ

---

## [Editor Report · Acceptance letter]

8 Jul 2024

PONE-D-24-15043R1 

PLOS ONE

Dear Dr. Ye, 

I'm pleased to inform you that your manuscript has been deemed suitable for publication in PLOS ONE. Congratulations! Your manuscript is now being handed over to our production team.

Kind regards, 

on behalf of

Dr. Zhaoqing Du 

Academic Editor

PLOS ONE